# A Deep Learning Approach for TUG and SPPB Score Prediction of (Pre-) Frail Older Adults on Real-Life IMU Data

**DOI:** 10.3390/healthcare9020149

**Published:** 2021-02-02

**Authors:** Björn Friedrich, Sandra Lau, Lena Elgert, Jürgen M. Bauer, Andreas Hein

**Affiliations:** 1Assistance Systems and Medical Device Technology, Carl von Ossietzky University, Ammerländer Heerstraße 114-118, 26129 Oldenburg, Germany; andreas.hein@uni-oldenburg.de; 2Geriatric Medicine, Carl von Ossietzky University, Ammerländer Heerstraße 114-118, 26129 Oldenburg, Germany; sandra.lau@uni-oldenburg.de; 3Peter L. Reichertz Institute for Medical Informatics of TU Braunschweig and Hannover Medical School, Carl-Neuberg-Straße 1, 30625 Hannover, Germany; lena.elgert@plri.de; 4Center for Geriatric Medicine, Agaplesion Bethanien Krankenhaus, University of Heidelberg, Rohrbacher Straße 149, 69117 Heidelberg, Germany; juergen.bauer@bethanien-heidelberg.de

**Keywords:** pre-frail, frail, older adults, mobility assessments, machine learning, supervised learning, decision support

## Abstract

Since older adults are prone to functional decline, using Inertial-Measurement-Units (IMU) for mobility assessment score prediction gives valuable information to physicians to diagnose changes in mobility and physical performance at an early stage and increases the chances of rehabilitation. This research introduces an approach for predicting the score of the Timed Up & Go test and Short-Physical-Performance-Battery assessment using IMU data and deep neural networks. The approach is validated on real-world data of a cohort of 20 frail or (pre-) frail older adults of an average of 84.7 years. The deep neural networks achieve an accuracy of about 95% for both tests for participants known by the network.

## 1. Introduction

With the demographic change a lot of challenges arise, particularly in the fields of medicine and healthcare. Older adults need frequent medical attention for maintaining their health and physical performance. Functional decline and mobility impairments are symptoms of impending diseases and also for frailty [1]. Early diagnosed diseases can be treated well, impairments can be minimised and there is a good chance of rehabilitation. Additionally, older adults have a high risk of falling and the medical implications can be more serious than the incident itself. Usually, standardised geriatrics assessments are used for monitoring the physical performance and estimating the risk of falling of older adults. Although digital solutions are available, the conventional assessments are still mainly performed under the supervision of a medical professional, therefore it is time consuming in clinical settings. Long-term monitoring on a regular basis assessments exceed the logistic capacities of medical professionals. Another disadvantage of assessments is the test situation itself, because people tend to give their best effort in test situations and studies showed that capacity is not performance [2,3,4]. Sensor assisted monitoring of the mobility and the physical performance in everyday life situations can give valuable information to health professionals for diagnosis and therapy. IMU sensors are low-cost and unobtrusive sensors for measuring the body’s specific force and angular acceleration. The light weight and the small size makes those sensors ideal for carrying around in a pocket or attached to a belt in daily life. Moreover, IMUs are not dependent on external infrastructure like satellites and can be used inside as well as outside without any loss of accuracy.

Machine learning approaches and recently deep learning showed good result in estimating gait parameters and fall risk of older adults [5]. One advantage of deep learning approaches is that the algorithms can approximate arbitrary functions, extract features automatically and the time consuming and difficult step of handcrafting features is not needed [6,7,8,9]. This advantage comes at the price of computationally expensive training and hyperparameter optimisation.

In this contribution a machine learning model for predicting the assessment scores for the Short-Physical-Performance-Battery (SPPB) and the Timed Up & Go test (TUG) on IMU data is introduced. The model learned from IMU data collected in everyday life situations from a cohort of older adults (84.75 y, 5.19 y SD). This article is structured as follows, in Section 2 the state of the art of mobility assessments and technology approaches for assisting are described. Technology assistance is divided in the two subgroups technology assisted assessments and unsupervised assessment approaches. In Section 3 the study for data acquisition, steps of preprocessing the data and the machine learning approach are explained. The results are shown in Section 4 and discussed in the following Section 5. In the last section conclusions are drawn and further steps are briefly mentioned.

## 2. State of the Art

Assessing the mobility of older adults is a common task in geriatrics medicine, validated and well accepted assessments like the SPPB [10] and the TUG [11] test are commonly used.

The TUG assesses the mobility of an older adult by getting up from a standardised chair, walking a distance of three metres, turning around and getting back to the chair to sit down. Assistive devices used for walking are permitted but it must be documented and used for any re-tests. The time from the start command “Go” until the patient’s buttocks touches the seat again is measured in seconds. Assessment categories are <10 s = normal, no mobility impairment, 11–19 s = minor mobility impairment not relevant in everyday life, 20–29 s = mobility impairment, >30 s severe mobility impairment, need for intervention [11]. Each category has numeric score, no mobility impairment is 1, minor mobility impairment is 2, mobility impairment is score 3, and severe mobility impairment is 4.

The SPPB consists of three parts assessing balance, gait speed and lower limb strength. During the balance test, the participant stands with the feet side by side, in semi-tandem stance and tandem stance for ten seconds each. Habitual gait speed is measured over a distance of 2.40 m, 3 m or 4 m. The chair rise test assess the muscle strength of the lower extremities by measuring the seconds the participant needs to perform five times from sit to stand. A maximum of 4 points for each task can be achieved. A total SPPB score ≤9 points was found as cut-off value for fit and frail people [10,12]. The approaches to support the assessments using technology can be categorised in technology assisted assessments and unsupervised assessments in real-life.

The technology assisted assessments are still performed in a way the assessment is supposed to be in order to use the validated point values for evaluation. Technical devices are used to enhance the measurements and to support the supervisors as well as the participants. In [13] an approach to enhance the TUG test was introduced. The measurements of the values’ score were computed and automatically measured using an IMU sensor. The computed values showed a high correlation to the gold standard measurement under supervision of a health professional using a stopwatch. A system for automated SPPB assessment executions has been developed in [14].

The unsupervised assessment approaches are trying to detect motion combinations from assessments in real-life by sensors. Once a motion combination is recognised the recorded sensor data is used to compute the performance. In [15] an approach for automatically detect TUG execution using an IMU sensor is shown and the results were compared to traditional stopwatch measurements. The TUG could be recognised with an accuracy of 96% and the results showed a strong correlation with the conventional method. As an item of the SPPB test and important parameter for functional decline, gait speed is focused as well. Approaches using cameras to measure the gait speed in domestic environments are introduced in [16,17,18]. Instead of cameras, more privacy respecting ambient sensors can be used as well like in [19,20,21,22]. However, all of those sensors are fixed and measure the gait speed in a single location only. To overcome this disadvantage small portable low-cost IMU sensors can be used to measure gait speed as well. The research in [23,24] showed, that gait speed estimations based on IMU data are comparable to gold-standard measurements of a GAITRite walkway. The experiments described in [25] showed the validity and reproducibility of using IMU sensors for gait parameter estimation. Besides the gait parameter measurements IMU data can be used to measure the intensity of activities [26].

Deep learning approaches showed good results on estimating gait parameters and the risk of falling on wearable sensor data [5]. Yu et al. used IMU data of Parkinson’s disease patients collected during TUG assessments for estimating the fall risk and the severity of Parkinson’s disease symptoms. The Convolutional Neural Network achieved an F-measure of 94% for estimating the fall risk and a Root-Mean-Squared-Error of 0.06 for severity estimation [27]. Similar approaches showed good results for people suffering from neurological disorders and multiple sclerosis. The networks achieved an accuracy of 92.1% and an Area Under the Curve (AUC) of 0.88 respectively [28,29]. Considering the parameters age and gender in addition to the raw sensor data significantly improves the performance of deep learning models [30]. Convolutional Neural Networks are also able to predict the frailty and cognitive dysfunction in respect to the mini-mental state examination of older adults. The network was trained with spectograms of walking-in-place data collected by IMUs. The network achieved an accuracy of 94.63% and 97.59% for frailty and cognitive dysfunction respectively [31]. An Artificial Neural Network in combination with a pressure sensor was used to detect abnormal foot postures. The pressure sensors were placed inside shoes and the Artificial Neural Network classified abnormal foot postures based on the gait characteristics with an accuracy of 99% [32]. Gait abnormalities can be detected using wrist-worn IMUs as well. A deep neural network trained IMU data collected by smartwatches achieved an accuracy of 88.90%, a sensitivity of 90.60%, and a specificity of 86.20% [33]. Deep Neural Networks also has been used to detect the fall incidents themselves. Using wearable sensor data an accuracy of 97.16% was achieved on the task of fall detection [34]. Using accelerometer data only a sensitivity of 88.20% and a specificity of 96.40% could be achieved in [35].

The mentioned approaches focus on enhancing the execution of geriatrics assessments using technology or detecting motion patterns from assessments in real-life. The approach in the article at hand is different, because the score of an assessment is predicted on IMU data. The participant neither has to execute any special motion patterns nor to complete the assessment in a certain place.

## 3. Methods and Materials

### 3.1. Data Acquisition

The data was collected during the observational OTAGO study in 2014 and 2015 over a period of 10 months [36]. The functional performance was assessed every month by the TUG and SPPB tests through conventional stopwatch measurements by a health professional. The cohort consisted of 20 participants (17 female, 3 male) aged 76 to 92 (mean 84.3 y, SD 5.19 y). At baseline, 14 participants (70%) were identified as frail (Frailty-Index ≥ 2 pts.) and six participants were pre-frail. The mean scores of the functional performance were 17.9 s for TUG and 5.95 points for SPPB. The baseline characteristics are presented in Table 1. Table 2 shows the characteristics of the cohort at the end of the study. The cohort size is reduced by two, because two participants deceased during the study. Despite dropout the available data of these two participants was considered. Each participant got one IMU sensor for data collection.

The IMU sensor of type Shimmer 3r was capable of measuring force in 9 Degrees of Freedom (9DOF) and was comprised of wide range and low range accelerometer, gyroscope, magnetometer, and pressure sensor [37]. In the first two weeks of the study the sensor was set to 51.2 Hz and in the remaining time of the study the sensor was set to 102.4 Hz. The sensors were given to the participants before the TUG and SPPB were done and collected after around about two weeks. Therefore, the dataset consists test situations and everyday life situations. Participants were asked to wear the the Shimmer3r IMU the whole day on a sensor belt, in a trouser pocket or other small pocket on the right side of the hip. The logo should face the front and the side with the charging port facing down. At night the sensor was supposed to be placed on the charging station. The participants were instructed to store the sensor safely during taking a shower and in the exercise bath. In total, 259 days of IMU data were collected.

### 3.2. Preprocessing

Before using the data for learning several preprocessing steps were applied. Participants 2, 3, and 4 were excluded, because of unavailable IMU data. The models are tested with two different testing strategies. The first testing strategy is the common testing strategy in machine learning, a test set is separated from the dataset. The second strategy is to test the models on data of participants which have been excluded before processing the data. The data of the excluded participants were neither used for training nor for evaluation of the model during training time. The participants 16 and 19 were randomly chosen for evaluating the performance of the model on unknown participants. This led to an exclusion of the SPPB score 2, because participant 19 was the only participant with a score of 2 in a SPPB assessment. Only two TUG assessments took longer than 30 s (max. +1.6 s). The scores were included in score 3, for not losing the data. The values of the low range accelerometer, wide range accelerometer, gyroscope and magnetometer were chosen as input to the network. The values of the temperature sensor were mostly 0 and contained errors and were not considered. The orientation in relation to the earth coordinate frame of the IMU was unknown and to eliminate the influence of the orientation the magnitude for each sensor modality was computed by
(1)$$m_{i}=\sqrt{x_{i}^{2}+y_{i}^{2}+z_{i}^{2}}$$
where *i* is the index of the value and x,y, and *z* are the axis of the sensor. After computing the magnitude the sensor data was filtered by a second order low-pass filter with a cut-off frequency of $\frac{1}{4}\times sampling\;frequency$. The filtered data has been divided into 5 s non-overlapping windows. Since two different sampling frequencies were used in the study the number of values per window differed. The number of values per window were rounded down to 500 values for the sampling frequency of 102.4 Hz and to 250 values for 51.2 Hz. Latter were oversampled to 500 values per window by duplicating each value. The final sets may contain data from assessments as well, but the amount of windows containing assessment data is insignificant. For each participant about 180,018.13 (SD ± 19,565.80) samples are available and a maximum of 1320 samples of each participants contain assessment data. The number of windows per class of the resultant dataset are shown in Table 3. Due to the setting of the data acquisition the classes were imbalanced and for several classes no windows were available at all. To balance the dataset windows of the overrepresented classes were deleted and the score 12 was excluded from the SPPB. Considering the smallest class as lower threshold for balancing, would have led to massive loss of data. The dataset was balanced two times and two different datasets, one for each assessment were derived. The dataset for the SPPB had 148,307 windows per class and the dataset for the TUG had 169,711 windows per class. The sets were divided into subsets for training (75%), validation (15%) and test (10%) in a stratified fashion.

The data of the two participants reserved for testing were preprocessed in the same way, but were not balanced and not divided into subsets. The data for SPPB score 2 of participant 19 were not considered for evaluation, because the model was not trained for that class. The number of values per class for participants 16 and 19 are shown in Table 4 and Table 5 respectively. The data was collected over a certain period of time and the physical performance changed during that time. So, one participant could have achieved different assessment scores.

Before the data was fed to the model the data was scaled to a range of 0 and 1.

### 3.3. Network Architecture

For this research a deep neural network approach was used. The architecture is shown in Figure 1 and the blocks are shown more detailed in Figure 2. The architecture and window size are adapted from [38]. Two models were trained, one for each assessment. The difference of the models was the number of neurons of the classification layer. For the SPPB model 9 neurons, and for the TUG model 3 neurons were used. In both networks the final classification layer is activated by the softmax function. The trained networks are available in the Appendix A.

As first layers Long-Short-Term-Memory (LSTM) layers are used for capturing the relation between the time steps. Then a convolutional layer was added to learn features for each sensor modality. All intermediate features were concatenated and forwarded to a sequence of convolutional layers. The final classification was performed by a two layer neural network. Dropout, maximum pooling and batch normalisation layers were added to prevent overfitting.

The model was trained using categorical cross-entropy as loss function and accuracy as metric. The accuracy was computed as follows
(2)$$accuracy=\frac{correct\;classified}{all\;samples}$$

The optimiser was AMSGrad version of the *Adaptive Moment Estimation* (Adam) with an initial learning rate of 0.001, a first order derivative momentum of 0.5, a second order derivative momentum of 0.8, and an exponential decay after the first 10 epochs [39].
(3)$$lr\left(epoch\right)=0.001\times e^{0.1\times (10-epoch)}$$

## 4. Results

The best epochs were epoch 67 for the SPPB model with a validation accuracy of 94.29% and epoch 52 for the TUG model with a validation accuracy of 95.89%. The accuracy on the test %set was 94.28%, and 95.79% for the SPPB and the TUG model respectively. The accuracy for the TUG scores (2, 3) for participant 16 was 98.84%, and 26.15% for participant 19. The accuracy for the SPPB scores (4 and 10) of participant 16 was 6.39%, and for SPPB scores (3, 4, 5, and 7) for participant 19 was 14.13%. The Table 6, Table 7 and Table 8 give an overview over the results. The Receiver Operating Characteristic (ROC) curves in Figure 3 and Figure 4 show high true positive and low false positive rates at high decision thresholds. The average ROC curves (blue) and the ROC curves for each class are showing similar progress and overlap. The AUCs of the TUG model and scores are 0.99 and the AUCs of the SPPB model are 1 except for the classes 7 and 9, where the AUCs are 0.99. The confusion matrices for the SPPB and TUG models are shown in Table 9 and Table 10. The SPPB model classified score 11 best with 573 false classifications (sensitivity: 96.14%, specificity: 99.27%) and score 9 worst with 1381 false classifications (sensitivity: 90.69%, specificity: 98.85%). The TUG model classified score 3 best with 771 (sensitivity: 96.44%, specificity: 97.93%) false classifications and score 2 worst with 1082 false classifications (sensitivity: 95.01%, specificity: 97.25%). The most false classification of the TUG model is for adjacent classes, e.g., 614 samples of class 1 were classified as class 2, but only 268 samples as class 3.

The Figure 5 and Figure 6 are showing the progress of the loss during training and the Figure 7 and Figure 8 are showing the accuracy during training. The validation loss and the validation accuracy are fluctuating in the beginning, but become stabilised after epoch 25. Overall, the loss and accuracy graphs showed the desired behaviour, increasing fast in the early epochs and stabilising during the later epochs. The graphs for both models are similar, but the SPPB model shows a little less performance and little higher loss than the TUG model.

The confusion matrices for participant 16 and 19 are shown in Table 11, Table 12, Table 13 and Table 14, respectively. The accuracy of the model is lower for the unknown participants. The accuracy for the class 2 of the TUG assessment is 98.93% and 99.49% for participant 16 and 19 respectively. The accuracy for the class 3 of the TUG assessment is 2.50% and 0.81% for participant 16 and 19 respectively. The class 1 is not available for the two participants. The best class for the SPPB score of participant 16 is class 4 with an accuracy of 12.51% and the best class for participant 19 is class 6 with an accuracy of 17.23%. The worst classes are 10 for participant 16 and 3 for participant 19 with an accuracy of 0.000641% and 0.00% respectively.

Participant 16 (76 y) was undergoing chemotherapy two months after the study began and deceased two months before the study ended. Therefore, values of only three assessment dates were available. Facing a severe loss of physical condition following inactivity, functional performance decreased very fast - especially in gait speed, but not in total scores of the SPPB. The interesting part is that, the intra-individual range varied from 4 to 10 points within the short period of time. TUG stopwatch measurements showed a more linear decline and a category change from 2 to 3.

The TUG results for participant 19 (90 y) were very close to the decision boundary of classes 2 and 3. The time of 3 of 6 TUG assessments were of an average of 0.39 s slower than the maximum of 19 s needed for scoring 2 points. The participant scored 3 in those assessments.

### Limitations

The cohort does not represent the complete scale of the SPPB and the scores 1, 2, and 12 are not learned by the model and hence the validated assessment is not completely represented by the model. Another point is the IMU, which was used by the participants independently. Using filtering approaches some invalid data was filtered, but certainly not all, e.g., if a participant would have given the IMU to another person, the invalid data of that person remains in the dataset. Inactivity covered by noise in the sensor signal will also remain in the dataset.

## 5. Discussion

The results showed that the models were performing well on all classes on known participants, but significantly worse on unknown participants. Finding one patient the network is not performing well for already proofs that the model cannot be used for unknown participants without further adjustments like fine tuning. The reason why the SPPB model performs slightly worse than the TUG model is that assessment has more classes and is more complex in execution than the TUG. Moreover, the score of balancing test of the SPPB is subject to the impression of the supervising professional, rather than an objective measurement. The models learned the variety amongst the participants of the OTAGO study, even though the cohort was very heterogeneous. The heterogeneity is expected to be the reason for the low model performance for the unknown participants 16 and 19. The TUG assessment results of participant 16 were very close to the decision boundary of two classes. This made it even more difficult for the model to distinguish the data of those classes for an unknown participant correctly. Considering Table 9, this seems to be a general problem of the TUG model. That shows the limitations of the assessment scores and boundaries. The scores are defined using full seconds, but with today’s technology much more precise measurements up to milliseconds are possible. The low accuracy on the SPPB data is mainly due to the exclusion from the training set, i.e. the participant is unknown to the network.

Even though the ROC curves, the AUCs and the accuracy show a high performance of the model, inferring reasons for score changes are not possible. A change in one item of SPPB changes the score of the assessment, so the change could be due to decreasing gait speed, a declining balance or decreasing lower limb strength. The same holds for the TUG test, because the same aspects are implicitly assessed. Standing up from a chair is dependent on the balance and lower limb strength, and the walking part assesses the gait speed. So, the TUG score incorporates the same dimensions like the SPPB.

The results showed that the models were performing well on all classes on known participants, but significantly worse on unknown participants. The loss and accuracy graphs showed the desired behaviour of increasing fast in the early epochs and stabilising during the later epochs. The graphs for both models were similar, but the SPPB model showed a slightly lesser performance and slightly higher loss than the TUG model. Since the SPPB assessment has more classes and is more complex in execution than the TUG, this is a reasonable finding. Moreover, the score of balancing test of the SPPB is subject to the impression of the supervising professional, rather than an objective measurement. The models learned the variety amongst the participants of the OTAGO study, even though the cohort was very heterogeneous. However, this heterogeneity is expected to be the reason for the low model performance for the unknown participants 16 and 19 as well.

Participant 16 (76 y) was undergoing chemotherapy two months after the study began and deceased two months before the study ended. Therefore, values of only three assessment dates were available. Facing a severe loss of physical condition following inactivity, functional performance decreased very fast - especially in gait speed, but not in total scores of the SPPB. The interesting part is that, the intra-individual range varied from 4 to 10 points within the short period of time. TUG stopwatch measurements showed a more linear decline and a category change from 2 to 3.

The TUG results for participant 19 (90 y) were very close to the decision boundary of classes 2 and 3. The time of 3 of 6 TUG assessments were on an average of 0.39 s slower than the maximum of 19 s needed for scoring 2 points. The participant scored 3 in those assessments. This made it even more difficult for the model to distinguish the data of those classes for an unknown participant correctly. The low accuracy on the SPPB data is mainly due to the exclusion from the training set, i.e. the participant is unknown to the network. The results show as well that the age is not important itself. Participant 16 is 14 y younger than participant 19, but the physical condition is much worse. The physical condition of participant 19 is clinically constant and varies slightly between two classes. The latter shows the limitations of the assessment categories and boundaries. The categories are defined using full seconds, but with today’s technology much more precise measurements up to milliseconds are possible.

## 6. Conclusions and Future Work

The results showed that it is possible to use machine learning to predict the geriatrics mobility assessment scores on real-life IMU data, even though the cohort was very heterogeneous and the IMUs were not rigidly attached to the body. The models performed well on known participants and were able to predict the scores of SPPB and TUG with an accuracy of 95.79% and 94.27% for the TUG and SPPB assessment respectively. The ROC curves, the specificities and the sensitvities for each class show, that the models performing well enough to be used by professionals. On the downside, the results showed that the models are very inaccurate on data of unknown participants.

The most promising approach is to use deep unsupervised learning. In the first step the network could be trained to approximate the underlying probability distribution of the training data. In the next step fine tuning for the new participant could be applied. Since one participant is likely to show only a certain subset of all available scores, the output of the network is the probability whether the sample belongs the class the network was fine tuned with. So, changes could be detected.

Another important point is the investigation of the performance on a different cohort. The cohort used for this research was very special, e.g., study inclusion criteria was being pre-frail at least. For healthier cohort of older adults an early detection of physical decline is important for early treatment and prevention as well. Long-term monitoring would be useful for those cohorts. So the models must be evaluated on data of a healthier cohort. Moreover, measures to simplify the model without loss of accuracy should be taken, because the data acquisition is difficult and costly. A less complex model needs less training data.

## Figures and Tables

**Figure 1 healthcare-09-00149-f001:**
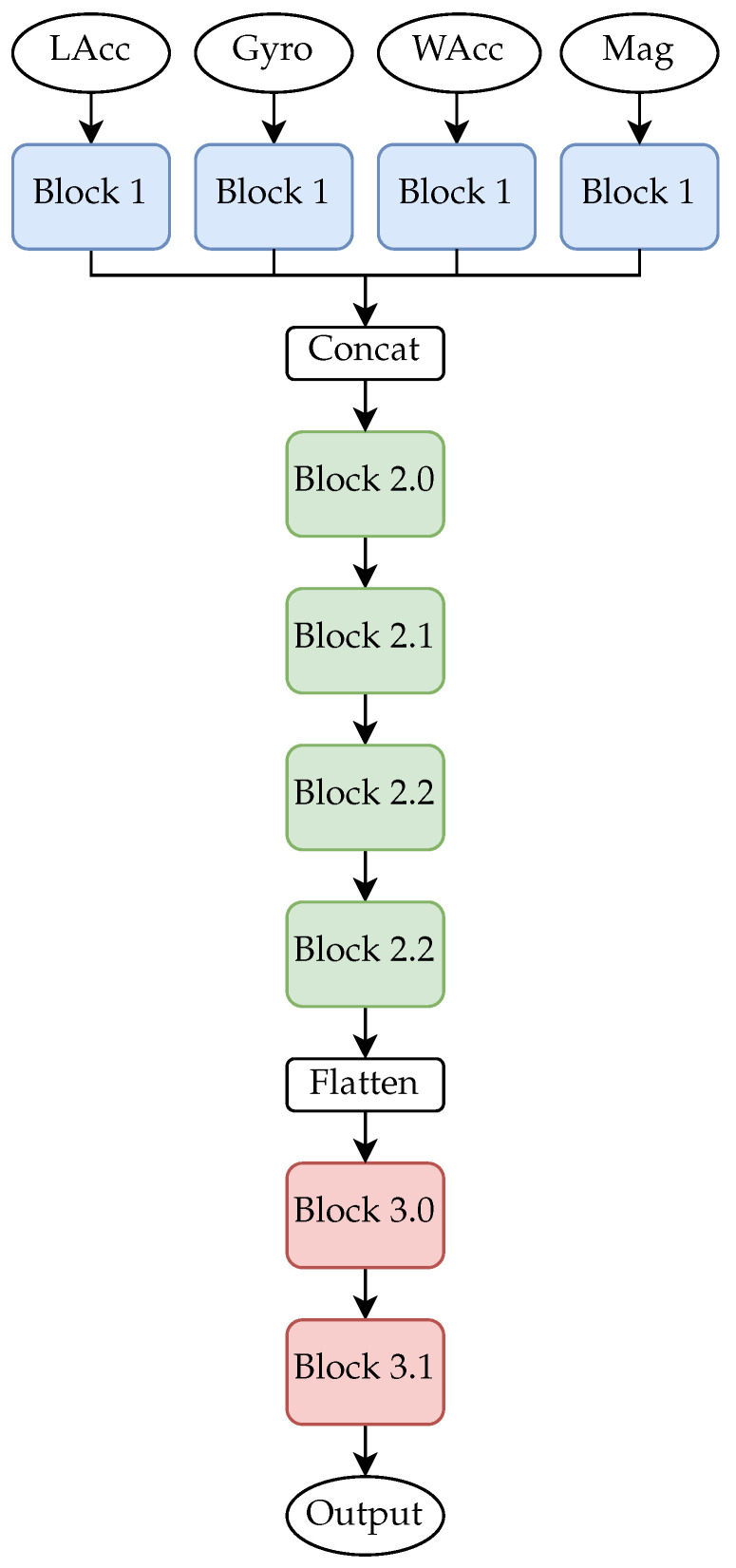
The deep neural network used for this research. Each sensor modality had its own input. The inner structure of the blocks are shown in Figure 2.

**Figure 2 healthcare-09-00149-f002:**
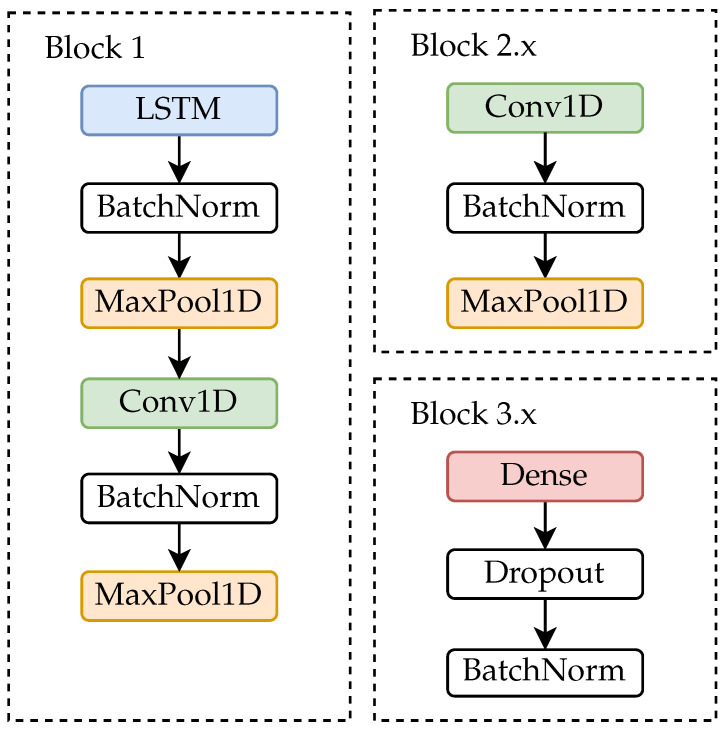
The blocks of the deep neural network. Details about the blocks and the layer parameters can be found in the Table A1, Table A2, Table A3, Table A4, Table A5 and Table A6 in Appendix B.

**Figure 3 healthcare-09-00149-f003:**
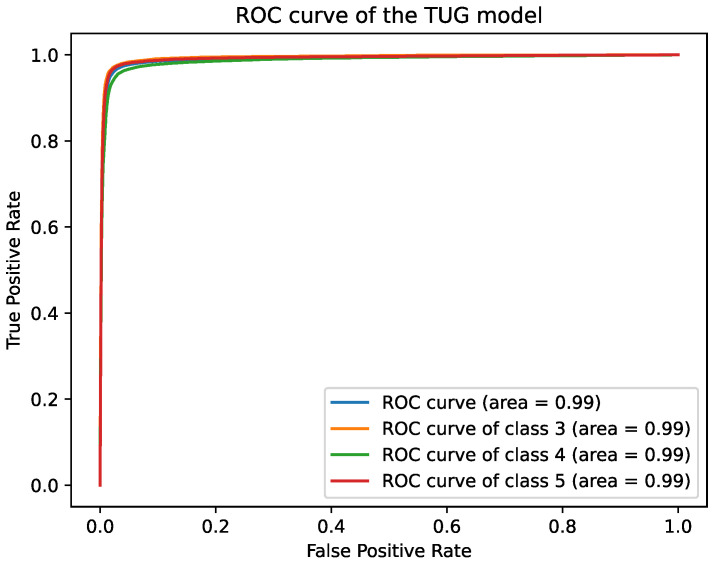
The ROC curve of the TUG model.

**Figure 4 healthcare-09-00149-f004:**
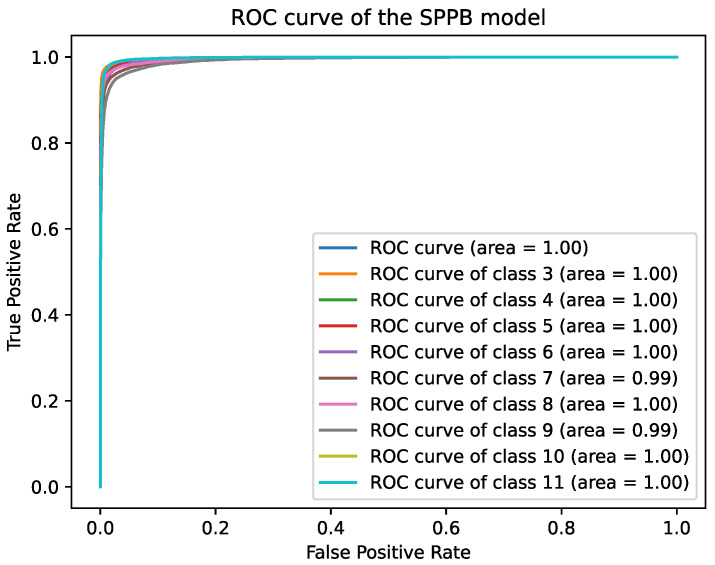
The ROC curve of the SPPB model.

**Figure 5 healthcare-09-00149-f005:**
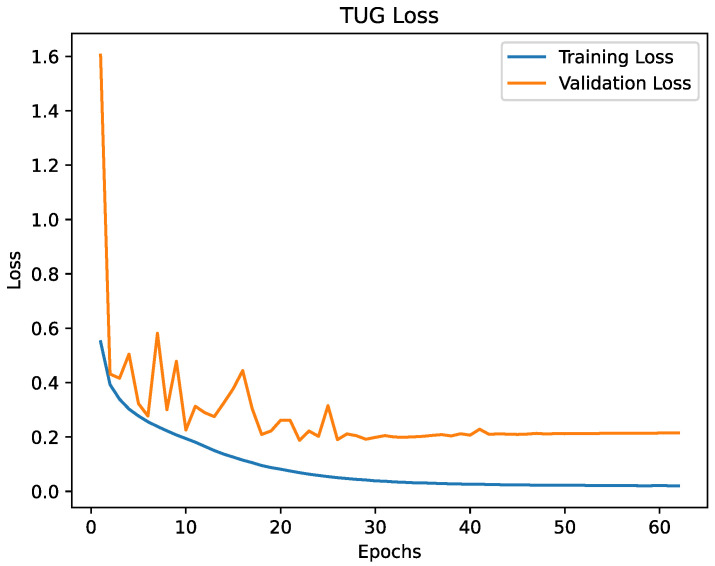
The loss of the TUG model. For the first 25 epochs the loss indicates that the learning rate is slightly too large. From epoch 25 the progress shows an asymptotic behaviour.

**Figure 6 healthcare-09-00149-f006:**
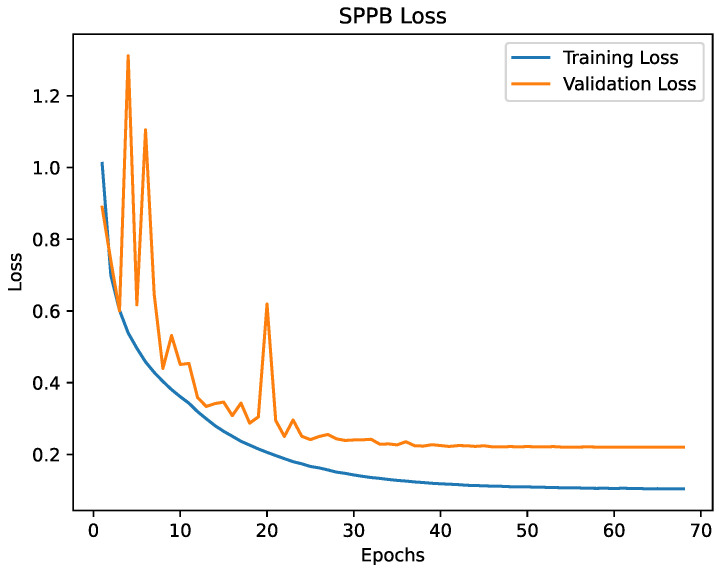
The loss of the SPPB model. For the first 25 epochs the loss indicates that the learning rate is slightly too large. From epoch 25 the progress shows an asymptotic behaviour.

**Figure 7 healthcare-09-00149-f007:**
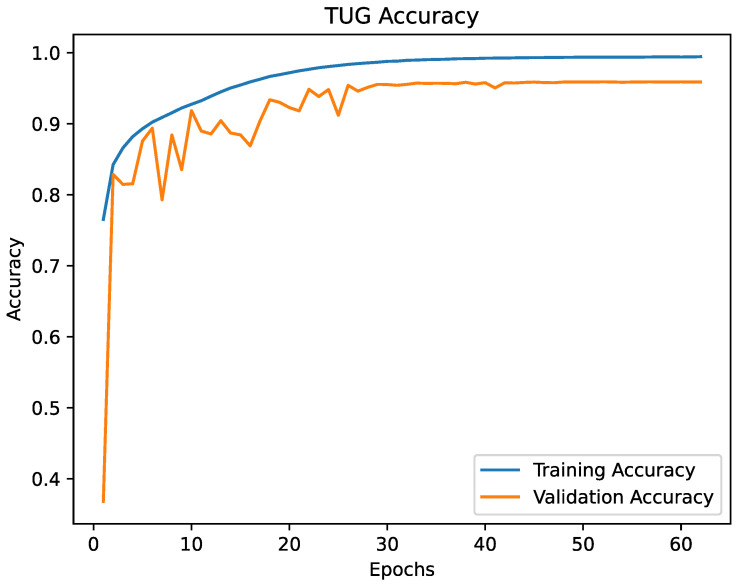
The accuracy of the TUG model. According to the progress of the loss, the accuracy fluctuates in the first 25 epochs and shows an asymptotic behaviour after epoch 25. The best validation accuracy score was 95.89% at epoch 52.

**Figure 8 healthcare-09-00149-f008:**
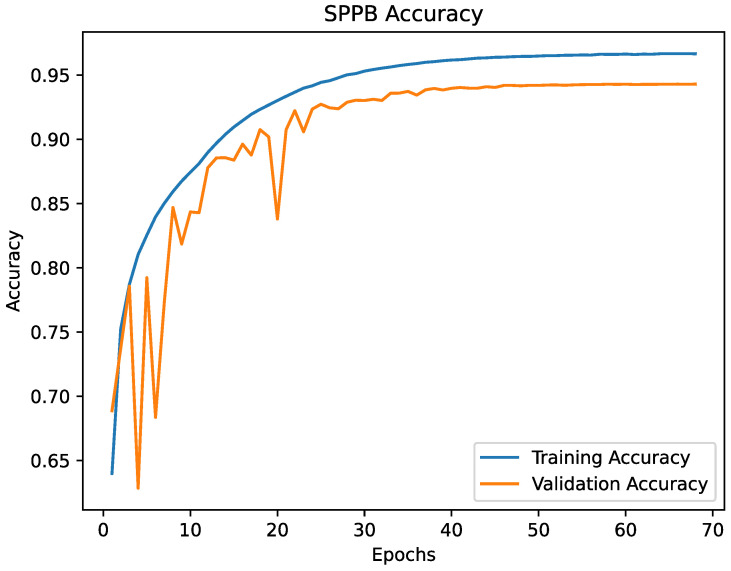
The accuracy of the SPPB model. According to the progress of the loss, the accuracy fluctuates in the first 25 epochs and shows an asymptotic behaviour after epoch 25. The best validation accuracy score was 94.29% at epoch 67.

**Table 1 healthcare-09-00149-t001:** The baseline characteristics of the study cohort.

*n* = 20	Age (y)	BMI ($\frac{\mathrm{kg}}{m^{2}}$)	Frailty Index (pts.)	SPPB (pts.)	TUG (s)
Mean	84.75	27.39	1.90	5.95	17.87
SD (±)	5.19	6.10	0.72	2.33	5.33
Range (min-max)	76.00–92.00	17.33–43.09	1.00–3.00	3.00–11.00	11.16 - 31.63

**Table 2 healthcare-09-00149-t002:** The characteristics at the end of the study cohort.

*n* = 18	Age (y)	BMI ($\frac{\mathrm{kg}}{m^{2}}$)	Frailty Index (pts.)	SPPB (pts.)	TUG (s)
Mean	85.44	28.27	2.00	6.61	16.12
SD (±)	4.92	6.44	0.97	2.85	5.85
Range (min-max)	77.00–93.00	16.89–45.99	0.00–4.00	2.00–12.00	8.15–30.06

**Table 3 healthcare-09-00149-t003:** This table shows the number of windows for each class of the SPPB and the TUG assessments. The range of the TUG score is smaller than the range of the SPPB score.

Score	TUG	SPPB
1	216,624	0
2	2,016,236	83,248
3	467,412	128,331
4	-	371,263
5	-	298,456
6	-	235,227
7	-	522,077
8	-	357,449
9	-	379,761
10	-	145,679
11	-	178,402
12	-	379

**Table 4 healthcare-09-00149-t004:** This table shows the number of windows for participant 16 who was excluded from the training set.

Score	TUG	SPPB
2	88,733	0
3	80	0
4	-	14,505
10	-	43,708

**Table 5 healthcare-09-00149-t005:** This table shows the number of windows for participant 19 who was excluded from the training set. Score 2 of the SPPB were not considered for evaluation, because the model for the SPPB was not trained with class 2.

Score	TUG	SPPB
2	28,336	256
3	82,010	8633
4	-	28,184
5	-	67,403
7	-	5868

**Table 6 healthcare-09-00149-t006:** The accuracy on the test set and the two excluded participants.

Assessment	Test Set	P16	P19
SPPB	94.27%	6.39%	14.13%
TUG	95.79%	98.84%	26.15%

**Table 7 healthcare-09-00149-t007:** The specificity and sensitivity for each TUG score on the test set.

Score	Sensitivity	Specificity
1	95.93%	98.50%
2	95.01%	97.25%
3	96.44%	97.93%

**Table 8 healthcare-09-00149-t008:** The specificity and sensitivity for each considered SPPB score on the test set.

Score	Sensitivity	Specificity
3	95.72%	99.67%
4	95.62%	99.38%
5	94.48%	99.45%
6	93.61%	99.06%
7	92.23%	99.25%
8	94.09%	99.31%
9	90.69%	98.85%
10	95.93%	99.31%
11	96.14%	99.27%

**Table 9 healthcare-09-00149-t009:** The confusion matrix of the TUG model. The class with the least false classifications is 1 and the class with the most false classifications is 3. t = true label, p = predicted label.

t\p	1	2	3
1	20,780	614	268
2	455	20,580	627
3	193	578	20,891

**Table 10 healthcare-09-00149-t010:** The confusion matrix of the SPPB model. The class with the least false classifications is 3 and the class with the most false classifications is 9. t = true label, p = predicted label.

t\p	3	4	5	6	7	8	9	10	11
3	14,195	98	35	147	80	95	123	47	10
4	39	14,181	44	163	111	40	157	69	26
5	27	67	14,012	205	72	76	187	155	29
6	99	153	136	13,883	107	116	140	107	89
7	61	154	70	118	13,677	123	229	86	312
8	73	43	52	154	121	13,954	188	97	148
9	62	144	169	173	211	189	13,449	221	212
10	13	46	117	88	47	73	180	14,226	40
11	12	25	24	73	136	106	155	42	14,257

**Table 11 healthcare-09-00149-t011:** The confusion matrix of the TUG for participant 16. The participant scores 2 and 3, but not 1. The most values are in class 2. Due to the imbalance the accuracy is high, but the performance is low. t = true label, p = predicted label.

t\p	1	2	3
1	0	0	0
2	8	87,782	943
3	1	77	2

**Table 12 healthcare-09-00149-t012:** The confusion matrix for SPPB of participant 16. The only class with correct predictions is class 4. t = true label, p = predicted label.

t\p	3	4	5	6	7	8	9	10	11
3	0	0	0	0	0	0	0	0	0
4	1814	5643	7484	4245	3043	9421	9322	4041	92
5	0	0	0	0	0	0	0	0	0
6	0	0	0	0	0	0	0	0	0
7	0	0	0	0	0	0	0	0	0
8	0	0	0	0	0	0	0	0	0
9	0	0	0	0	0	0	0	0	0
10	3921	15	2977	814	7112	22,327	4832	28	1682
11	0	0	0	0	0	0	0	0	0

**Table 13 healthcare-09-00149-t013:** The confusion matrix for TUG of participant 19. The model is not able to distinguish between score 3 and score 2 for this participant. t = true label, p = predicted label.

t\p	1	2	3
1	0	0	0
2	3	28,191	142
3	86	81,259	665

**Table 14 healthcare-09-00149-t014:** The confusion matrix for SPPB of participant 19. This participant was the only one with a SPPB score of 2. Since the model was not trained to classify this score, the values for score 2 were not considered for classification. t = true label, p = predicted label.

t\p	3	4	5	6	7	8	9	10	11
3	197	1	37	1	13	3	0	1	3
4	39	1,756	11	235	1753	328	403	99	4009
5	28	14	19,027	299	6143	444	1255	778	198
6	2276	219	15,315	974	363	2388	14,474	30,534	860
7	0	0	0	0	0	0	0	0	0
8	8	594	70	267	220	398	4234	75	2
9	0	0	0	0	0	0	0	0	0
10	0	0	0	0	0	0	0	0	0
11	0	0	0	0	0	0	0	0	0

## Data Availability

The data are not publicly available due to privacy concerns.

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
