# Peer review of "A Deep Learning Approach for TUG and SPPB Score Prediction of (Pre-) Frail Older Adults on Real-Life IMU Data"

_healthcare, 2021, doi:10.3390/healthcare9020149_

Round 1

Reviewer 1 Report

The authors addressed all my concerns. The introduction, discussion and results have been suitably improved.

Author Response

Dear respected reviewer,

thank you. We appreciate your effort and your comments and suggestions greatly increased the quality of the article.

Kind Regards

Björn Friedrich

Reviewer 2 Report

Authors have addressed almost all my previous comments.

However, some of the points clarified in the responses should be included in the text.

In addition, it is not still clear why they do not use k-fold validation to test the system. If this procedure is not followed, obtained results are completely dependent on the particularities of the users employed for testing. More tests should be performed to average the results.

The computational cost of a k-fold validation cannot justify the fact that a more rigorous method should be employed to validate the proposal.

Author Response

This manuscript is a resubmission of an earlier submission. The following is a list of the peer review reports and author responses from that submission.

Round 1

Reviewer 1 Report

The subject of the study carried out by the authors is of interest and contributes to the state of the art of the research field. However, it is necessary to improve the manuscript by providing further development of several sections, as well as modifying various aspects related to the way in which the information and results are presented. I indicate them below:

  • The introduction section is short and lacks information. It is necessary to deepen in the context of gait analysis. Machine Learning and especially Deep Learning have contributed to this field and there are several recent studies focused on the identification of events and pathologies directly related to gait, such as [1-5]. Contemplating this quote and other related ones allows a better justification about the focus of the study.
  • The figures used are too large and make reading difficult. I suggest the edition to better fit the structure of the journal's publications. It would be necessary to modify the diagrams to make better use of the space on the page.
  • The values for each cell of the confusion matrices should be expressed in percentages, not in absolute values.
  • I suggest the inclusion of ROC curves and a discussion of the results obtained.
  • The discussion must be expanded by adding a more complete analysis of the different results obtained. I suggest combining the results and discussion sections and exposing the authors' ideas and interpretations by looking at each result (or set of results) shown.

[1] Luna-Perejón, Francisco, et al. "Low-Power Embedded System for Gait Classification Using Neural Networks." Journal of Low Power Electronics and Applications 10.2 (2020): 14.

[2] Musci, Mirto, et al. "Online Fall Detection using Recurrent Neural Networks on Smart Wearable Devices." IEEE Transactions on Emerging Topics in Computing (2020).

[3] Ma, Liang, et al. "Room-level fall detection based on ultra-wideband (UWB) monostatic radar and convolutional long short-term memory (LSTM)." Sensors 20.4 (2020): 1105.

[4] Luna-Perejón, Francisco, Manuel Jesús Domínguez-Morales, and Antón Civit-Balcells. "Wearable fall detector using recurrent neural networks." Sensors 19.22 (2019): 4885.

Reviewer 2 Report

The topic is of great interest. However, the paper presents some technical and formal flaws that should be corrected before publication.

The use of deep learning to assess frailty of elderly people or FRA (fall risk assessment) is not new. The SoA can be clearly improved. I suggest revising the works of other authors, such as:

-Walking-in-Place Characteristics-Based Geriatric Assessment Using Deep Convolutional Neural Networks, by D.Jung et al.

- Motion Sensor-Based Assessment on Fall Risk and Parkinson's Disease Severity: A Deep Multi-Source Multi-Task Learning (DMML) Approach by S. Yu et al.

- Deep Learning to Predict Falls in Older Adults Based on Daily-Life Trunk Accelerometry

by Ahmed Nait Aicha

In this sense, an interesting review on this issue is presented in ‘Fall risk assessment in the wild: A critical examination of wearable sensors use in free-living conditions’ by M. Nouredanesh et al.

-The total duration (minutes or seconds) of the employed mobility samples (before using the windowing technique) should be indicated.

-The election of the observation window should be discussed or supported by other studies? Why 5 s? Why non-overlapping windows?

-I guess that elderly people spend a lot of time on a chair. Do the training sets include all the samples captured by the sensor during an (almost) completely quiet position?

-Why using two particular individuals for testing? Why not using a k-fold validation (to average the performance metric when the sets for training and set are changed)?

-Two persons are too few to test a classification system. All the scores are not represented in the test set.

-Table 4 (or 5). I am not an expert on TUG or SPPB tests, but how can the same user exhibit two different scores for the same test? This should be clarified. Which is the relationship between the TUG score and the categories described in lines 45-47?

-The CNN is poorly described. The dimensions of the architecture and hyper-parameterization (number of blocks, number of convolutional layers of each block, etc.) of the deep neural network should be commented and justified in more detail. The meaning of some blocks (e.g. Dense) should be clarified

-Which techniques are used to prevent overfitting?

-It is not clear which classification algorithm is employed in the last stage to produce the final score prediction.

-Comment if any pre-processing or normalization is applied to the input signals.

-Specificity and sensitivity of a supervised method should be measured against the test set (and not against the whole dataset). I guess that Tables 7 & 8 present the results for the whole dataset as they include the sensitivity and sensibility for scores that are not present in the individuals selected for the test set.

Other aspects:

-Page 3. Line 84: A reference to OTAGO project is recommendable. Are these data publicly available?

A reference to SHIMMER sensor official page should be provided.
